# Changes in Mental Health and Views on Communication and Activities of Public Institutions among Swedes during the COVID-19 Pandemic—A Cross-Sectional Repeated Measures Design

**DOI:** 10.3390/healthcare9111498

**Published:** 2021-11-03

**Authors:** Birgitta Kerstis, Fabrizia Giannotta, Petra von Heideken Wågert, Charlotta Hellström, Daniel Lindberg, Jonas Stier, Maria Elvén

**Affiliations:** 1School of Health, Care and Social Welfare Mälardalen University, 72123 Västerås, Sweden; fabrizia.giannotta@mdh.se (F.G.); petra.heideken.wagert@mdh.se (P.v.H.W.); charlotta.hellstrom@mdh.se (C.H.); daniel.lindberg@mdh.se (D.L.); jonas.stier@mdh.se (J.S.); maria.elven@mdh.se (M.E.); 2Department of Public Health Sciences, Stockholm University, 10691 Stockholm, Sweden

**Keywords:** age and sex differences, COVID-19, mental health, pandemic, public institutions, Sweden

## Abstract

Although many studies have been conducted on the effects of COVID-19 on individual lives, only a few focus on the changes in mental health and views of public institutions during the pandemic. This study aimed to investigate how mental health, i.e., life satisfaction, worries, and psychological distress, and views on public institutions’ communication and activities have changed among Swedes during the COVID-19 pandemic, and whether this was moderated by age and sex. In April–May 2020 (survey 1) and in January–February 2021 (survey 2), 2554 adults and 1904 newly recruited adults, respectively, anonymously completed online surveys. We found that life satisfaction and psychological distress did not change from survey 1 to survey 2. However, the level of worries increased, and the positive views of the public institutions decreased. Moreover, worries and psychological distress increased more in young adults than older adults. Finally, the change in the views of the public institutions was not related to the change in worries. Our results highlight the COVID-19 long-term impacts on individual mental health and call for the need for future research concerning the consequences for the population, especially among young adults. The results also indicate that the views on activities of public authorities decreased over time, especially among men. Given that loss of this trust can have vastly negative effects, for instance, on the vaccine campaign, it is important to monitor this trend, to increase awareness among Swedish authorities. The results also stress for institutions to provide adequate support both during the COVID-19 pandemic and in a future crisis.

## 1. Introduction

The World Health Organization (WHO) declared Coronavirus disease 2019 (COVID-19) as a global pandemic in March 2020. Individuals of all ages have experienced, to a different extent, negative consequences of the COVID-19 pandemic. For instance, even though young adults have not suffered much from the lethality of the virus [1], preventive measures, such as the closing of universities and restrained social contacts, in conjunction with a deteriorated economic situation in many countries, including Sweden [2], are likely to negatively impact their mental health. Moreover, as many adults and children are working and studying from home, there can be an increased risk of strained relationships and conflicts [3]. COVID-19 also has negative consequences for the oldest population, as they are the group more at risk to die from the virus [4], and their social life has been dramatically changed because of the pandemic. For instance, in Sweden, where this study took place, measures, such as the shutdown of community organizations, dramatically limit the social interactions of the older population (70 years and older) [5]. In addition, the lack of digital literacy among many of them (especially in the group 76 years and older) might have compromised both the access to public information [6] and the maintenance of a social life, for instance, through the internet. To sum up, in one way or another, the pandemic is likely to negatively impact the life and the wellbeing of all individuals, and not only due to its medical-related effects. Therefore, it is important to investigate all the possible negative consequences of COVID-19 to plan preventive measures to contrast the pandemic’s negative effects.

While many studies have highlighted an increase in mental health problems, especially in women and young adults during the beginning of the pandemic [7], studies that follow up the effects of the pandemic in later stages are still scarce. Among the few, a longitudinal study from the UK described an overall increase in mental distress in a cohort aged 16 years and older in April 2020, measured before and during COVID-19 [8]. Conversely, a study from the US revealed a decrease in psychological distress in the general population from April–June 2020 to March–April 2021 [9]. These contrasting results call for further investigation on the short- as well as long-term effects of the pandemic on the mental health of the individuals, especially as measures in Sweden were different than those used in many parts of the world.

In Sweden, preventive measures combating the COVID-19 contagion have been mostly based on voluntary and stepwise actions [10] instead of legislation and compulsory measures, including lockdowns, as in many countries. One of the rationales for this approach is that Swedish society is built on mutual trust and individual responsibility [11]. During a crisis, trust both between people and between people and the government is a key factor [12]. Therefore, a decrease in this trust might be particularly deleterious. Nevertheless, the higher number of deaths in Sweden compared with other Scandinavian countries that implemented lockdown strategies [13] might have influenced the populations’ views on public institutions. As the Swedish approach to the pandemic partly depends on the trust toward the communication and activities of public institutions, it is important to investigate whether and how the people’s views regarding this have changed over the year affected by the pandemic.

A relationship between information about COVID-19 and anxiety and stress symptoms [14] has been found. A Spanish study pointed out that having reliable and correct information decreased anxiety [15]. However, a decrease in trust toward the public institutions can also affect individual mental health with increasing worries and psychological distress [16]. Therefore, there is a need to investigate, in relation to age and sex, whether and to what extent individual mental health has changed and whether this was related to a change in trust towards the public institutions.

The consequences of the pandemic seem to affect population mental health in different ways, depending on age and sex [17]. Studies during the beginning of COVID-19 demonstrate that women and young people seem to be more affected in terms of mental health compared with men and other age groups [18,19]. However, it is unclear if these differences are persisting. Hence, there are reasons to explore if and how the pandemic has impacted different age groups and sexes concerning both mental health and trust toward public institutions.

In this study, we focus on some aspects that can indicate good mental health, such as good life satisfaction, low worries, and psychological distress [20].

To sum up, the present study aims to explore whether:(1)Mental health, i.e., life satisfaction, worries, and psychological distress, and views on communication and activities of public institutions have changed among Swedes during the pandemic;(2)This change is moderated by age and sex;(3)Changes in the views on communication and activities of public institutions are related to changes in mental health.

## 2. Materials and Methods

### 2.1. Study Design

The study used a descriptive cross-sectional repeated measures design and was part of a World Values Survey 3-measure “Values in Crisis” survey, studying peoples’ values during the COVID-19 pandemic [21].

### 2.2. Participants and Setting

The first survey, hereafter named survey 1, was conducted in April–May 2020, and the second survey, hereafter named survey 2, was conducted in February–March 2021. A total of 40,000 potential participants, randomly recruited from the Swedish population between 18 and 79 years of age, were available through a survey management service. Of these 40,000, a stratified sample of 4561 representing the Swedish population were contacted and asked in survey 1. Of these 4561, 2554 participants anonymously completed the first online survey, yielding a 56% response rate. During survey 2, among the same 40,000 potential participants, 2539 were newly recruited anonymously. Of these 2539, 1904 completed the online survey, yielding a 75% response rate. The study participants were introduced to the purpose of the study and provided with an information letter and a link to a self-administered 20 min online questionnaire, including a total number of 150 items, of which x items were analyzed in the current study. The participants were informed that they consented to be included in the study by answering the web-based questionnaire, for which they received a cinema ticket. The study was conducted in accordance with the Declaration of Helsinki [22] and Swedish law [23]. All personal data connections were deleted after the material was collected and were not accessible to the researchers in the present study.

### 2.3. Measures

#### 2.3.1. Life Satisfaction

The participants self-estimated their life satisfaction with the question: All things considered, how satisfied are you these days with your: (1) Health condition, (2) Financial situation, (3) Social relations, (4) Work-life balance, and (5) Life as a whole under the given circumstances (i.e., COVID-19). The response options ranged from Completely dissatisfied (1) to Completely satisfied (10). An index of life satisfaction was developed where high values indicated high satisfaction. Cronbach’s alpha was 0.84 in survey 1 and 0.85 in survey 2.

#### 2.3.2. Worries

The participants self-estimated: How do you rate your concern regarding the current Corona outbreak? How worried are you about: (1) Getting the virus; (2) Risk to infect others without knowing; (3) Need to change lifestyle for good, and (4) That the health care system does not have enough resources. The response options ranged from Very worried (1) to Not at all worried (5), with a maximum score of 20. The items were reversed, and an index was developed where high values indicated higher worries related to COVID-19. Cronbach’s alpha was 0.62 in survey 1 and 0.64 in survey 2.

#### 2.3.3. Psychological Distress

The participants self-estimated: Over the last two weeks, how often have you been bothered by the following problems: (1) Feeling nervous, anxious, or on edge, (2) Not being able to stop or control worrying, (3) Feeling down, depressed, or hopeless, (4) Little interest or pleasure in doing things, and (5) I have felt lonely. The response options ranged from Not at all (1) to Nearly every day (4). The items were combined in an index where higher scores indicated a higher level of psychological distress. Cronbach’s alpha was 0.84 in survey 1 and 0.86 in survey 2.

#### 2.3.4. The Views on Public Institutions’ Communication and Activities

The participants self-estimated: How do you view the following activities and communication concerning the COVID-19 outbreak concerning seven public institutions (i.e., the health care service, the municipality, the National Board of Health and Welfare, the politicians, the public health agency, the public media, and the Swedish agency for social protection and preparedness). The response options ranged from Very good (1), Good (2), Bad (3), Very bad (4), to Don’t know (5). The response option Don’t know was treated as missing. The items were reversed, and an index was developed where higher values indicated a positive view. Cronbach’s alpha was 0.89 in survey 1 and 0.88 in survey 2.

### 2.4. Statistical Analysis

Descriptive statistics were presented using frequencies and percentages for categorical variables and means and standard deviations for continuous variables. To investigate the changes in life satisfaction, worries, psychological distress, and views on public institutions’ communication and activities related to COVID-19, *t*-tests for independent samples were used.

To test the moderation hypothesis, we used a series of multiple linear regression models where survey number, age, sex, and the interaction terms were entered as independent variables [24]. In all the models, age was centered to reduce collinearity problems. Finally, to test whether a change in views on public institutions’ communication and activities was related to a change in mental health, a multiple regression model was applied, entering survey number and views on public institutions’ communication and activities and the interaction between those two as independent variables, after centering public institutions’ communication to reduce multicollinearity. All the variables that were treated as continuous hold an acceptable level of skewness and kurtosis. All tests were two tailed, and the statistical significance was set at *p* ≤ 0.05. All analyses were performed using IBM SPSS Statistics (version 26.0; IBM SPSS, Armonk, NY, USA).

## 3. Results

### 3.1. Sample Description

The socio-demographic characteristics for the sample in its entirety and for the two surveys are presented in Table 1. The mean age of the whole population was 51.2 years (SD = 16.9); in survey 1, it was 49.3 years (SD = 17.1), and in survey 2, it was 52.4 years (SD = 16.6). The whole sample included 52.5% women; in survey 1, 52.0% were women, while in survey 2, 53.1% were women. The two samples were comparable and differ only by age and occupation rate (Table 1).

### 3.2. Changes in Mental Health and Views on Communication and Activities of Public Institutions during the Pandemic

There were no differences in mean levels of life satisfaction and psychological distress. The level of worries increased and the views on public institutions’ communication became less positive in survey 2 compared with survey 1 (Table 2).

#### Moderation Effects of Age and Sex

Life satisfaction did not change between the surveys and was not affected by sex or the interaction between age and sex. Older age predicted better life satisfaction. The level of worries increased from survey 1 to survey 2. Being women and older adults predicted higher levels of worries. Sex did not moderate the increase in worries while age did. The increase in levels of worries was higher in the young adults compared to the older age groups (Table 3 and Figure 1). Psychological distress did not change between the surveys and was not affected by age or sex (Table 3). However, an interaction effect with age was found. Specifically, psychological distress increased from survey 1 to survey 2, but only in young adults (Table 3 and Figure 2).

The views on public institutions’ communication and activities became more negative from survey 1 to survey 2 and were predicted by age and sex. Being women and older adults predicted a more positive view of the public institutions. Finally, when it comes to the change between survey 1 and survey 2 (Table 3), the decrease in positive views of the public institutions did not differ between young and older adults, while positive views of the public institutions decreased more in men than women (Table 3 and Figure 3).

### 3.3. Relation between Changes in Views of Public Institutions’ Activities and Worries in the Surveys

We used a multiple regression model with an interaction term (measure*views on public institutions’ activities) to investigate whether the change in the views of public institutions was related to the change in worries (F (3; 4323) = 18.14, *p* < 0.001). However, the changes in worries were not related to the change in the public view of public institutions (*β* = −0.039, *p* = 0.45).

## 4. Discussion

This study investigated how and whether mental health and views on communication and activities of public institutions changed during the COVID-19 pandemic in Sweden, and whether this change was moderated by age and sex. Our results indicate that the average levels of life satisfaction and psychological distress were not affected by the pandemic and did not change from survey 1 to survey 2. However, there was an increase in worries about the pandemic and a decrease in the trust in public institutions’ communication and activities. This may suggest that, at the beginning of the pandemic, the Swedes judged that the pandemic was going to be only for a short time, and it was hard to predict its difficult long-term consequences. About nine months later, when survey 2 was conducted, the consequences of the pandemic were part of many people’s daily life, affecting their worries.

When it comes to the views on public institutions’ communication and activities, we observed a general decrease in positive attitudes toward activities and communication from the public institution for all populations. This change was even more accentuated in men compared with women. This is in contrast with another study that described an increase in trust towards the public institutions during the first outbreak in Sweden [25]. Swedes, in general, have a high level of confidence and trust in the government and authorities [26]. In addition, the Swedish government has adhered to recommendations rather than regulations [11], especially during the first months of the pandemic. This might have influenced the views of the institutions at the beginning. However, the tightening of the restrictions to contrast the pandemic during survey 2 and the change in the so-called Swedish strategy might have affected the opinion of the population regarding the activities of the government. Nevertheless, the changes were not associated with the change in individual mental health, i.e., worries; therefore, we cannot establish that the restrictions decided by governments are the critical factor in explaining increased worry or mental distress in general. However, the change of trust in institutions might have other consequences for the population. For instance, a recent study describes a link between trust in public institutions and the adoption of the recommended preventive measures to contrast the spread of the virus [27]. Therefore, the consequences of a change in views of the public institutions need to be explored further.

The second aim was to investigate the role of sex and age in moderating the change in mental health and views on the communication of the institutions from survey 1 to survey 2. We found that sex moderated only the effects on views of the public institutions, making the decrease in positive views even worse for men than women. However, age also showed a stronger moderating effect. The young adults had the highest increase in worries. Moreover, their psychological distress was increased while the other age groups were stable. This indicates that young adults might have a stronger impact on mental health compared with other age groups. This is consistent with studies describing that young people’s mental health was negatively affected during COVID-19, during the first outbreak of the pandemic [18,19]. This study adds that this deterioration of mental health persists nine months later, suggesting that it might become a phenomenon of concern.

The reasons why young adults seem to be more psychologically affected by the pandemic are still unknown. However, there are some hypotheses. In Sweden, the restrictions were few during the first months but became stricter during autumn 2020, which might have especially affected young adults, as bars and restaurant opening hours were limited. Furthermore, for young adults, the pandemic has made access to professional and friend support more difficult [28]. In addition, the lockdown of universities has affected young adults’ education and work opportunities as well as possibilities to gain life experiences [29]. All these changes might have affected the mental health of young adults. Further studies are needed to confirm these hypotheses.

### Limitations and Strengths

A strength of the present research was that it was possible to examine the role played by a range of mediating factors have had in explaining longitudinal changes in distress during the COVID-19 pandemic among a large portion of the population.

This study has some weaknesses. Although the relatively large population implies good statistical power, some limitations should be acknowledged. First, the cross-sectional design with two different samples precludes assumptions about temporality or causality. Moreover, the psychological distress measure was not directly referred to COVID-19, and there is no information about the participants’ status before the pandemic. Therefore, we cannot exclude that the most psychologically distressed people were already so before the pandemic. Another limitation is the high no-response rate, especially during survey 1; it could be that the people not answering are the ones suffering the most with personal consequences. Furthermore, well-educated people and women were overrepresented in the sample. All these effects might affect the representativeness of the sample. Nevertheless, this no-response rate of our study is quite in line with most population studies. Moreover, the internal reliability of the worries measure was relatively low but still acceptable Finally, the results of the present study might not be generalizable to other countries, populations, or age groups, although the use of a relatively large sample from the general population allows for broad generalizability to similar contexts. Therefore, longitudinal studies following a cohort and with a more representative sample are needed.

This study has some strengths too. The most important one is that this is one of the few studies to investigate the changes in both mental health and views of the public institutions during COVID-19, using a population sample that ensures statistical power and makes the results reliable.

## 5. Conclusions

Many studies have been conducted during COVID-19, but only a few focus on changes in mental health and views of public authorities during the pandemic. Moreover, our study highlighted that age, especially, but also gender, affected these changes. Given the negative impact of the pandemic on young adults’ mental health which emerged from our study, we conclude that it is especially important to promote mental health among young adults as they will work and serve our society for many years, hopefully with good health and trust in public institutions. The results also indicate that the views on activities of public authorities decreased over time, especially among men. Given that loss of this trust can have vastly negative effects, for instance, on the vaccine campaign, it is important to monitor this trend, to increase awareness among Swedish authorities. The results also stress for institutions to provide adequate support both during the COVID-19 pandemic and in a future crisis.

## Figures and Tables

**Figure 1 healthcare-09-01498-f001:**
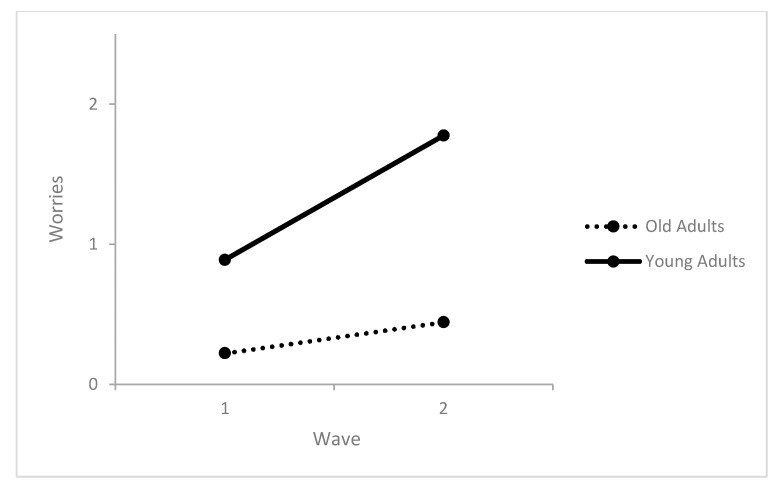
Interactions between the number of surveys and age predicting worries.

**Figure 2 healthcare-09-01498-f002:**
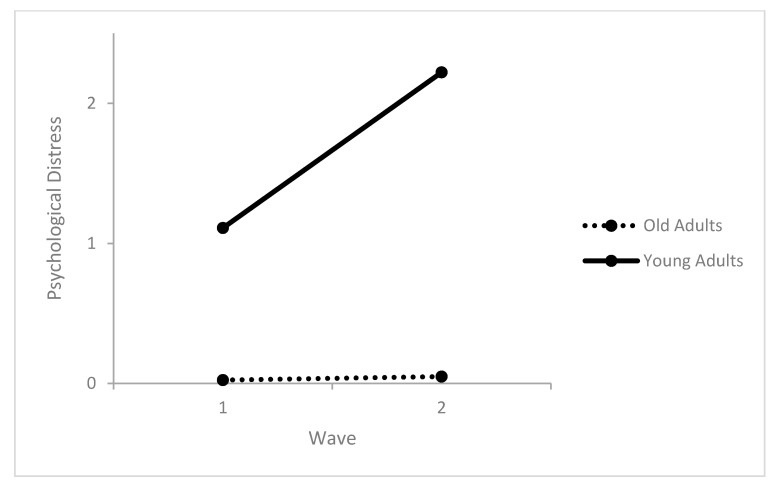
Interactions between the number of surveys and age predicting psychological distress.

**Figure 3 healthcare-09-01498-f003:**
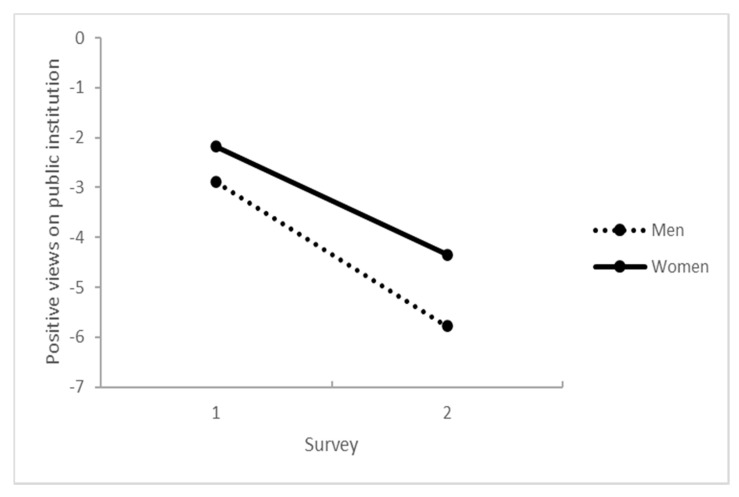
Interactions between the number of surveys and sex predicting positive views on communication and activities of public institutions.

**Table 1 healthcare-09-01498-t001:** Demographic characteristics for the two surveys.

	Total *N* = 4458	Survey 1 *N* = 2554	Survey 2 *N* = 1904	*p*-Value between Surveys
Age mean (SD)	51.2 (16.9)	49.3 (17.1)	52.44 (16.6)	<0.001
Sex *n* (%)				
Men Women	2118 (47.5) 2340 (52.5)	1225 (48.0) 1329 (52.0)	893 (46.9) 1011 (53.1)	0.999
Highest education *n* (%)				
Compulsory school (9 years) Senior high school University Other	237 (5.3) 1751 (39.3) 2461 (55.2) 9 (0.2)	139 (5.4) 1013 (39.7) 1396 (54.7) 6 (0.2)	98 (5.1) 738 (38.8) 1065 (55.9) 3 (0.2)	0.406
Occupation *n* (%)				
Students Manual workers Non-manual workers Self-employed Parental leave Sick leave/early retired Retired Unemployed Other	395 (8.9) 1019 (22.9) 1547 (34.7) 269 (6.0) 53 (1.2) 119 (2.7) 928 (20.8) 73 (1.6) 55 (1.2)	254 (9.9) 597 (23.4) 872 (34.1) 151 (5.9) 35 (1.4) 68 (2.7) 504 (19.7) 44 (1.7) 29 (1.1)	141 (7.4) 422 (22.2) 675 (35.5) 118 (6.2) 18 (0.9) 51 (2.7) 424 (22.3) 29 (1.5) 26 (1.4)	0.012
Cohabitation status *n* (%)				
Living with partner	3041 (68.4)	1729 (67.7)	1312 (69.0)	0.174
Origin *n* (%)				
Born in Sweden Mother born in Sweden Father born in Sweden	4105 (94.2) 3963 (91.0) 3785 (86.9)	2347 (94.4) 2269 (91.2) 2164 (87.0)	1758 (94.1) 1694 (90.6) 1621 (86.7)	0.927
Housing *n* (%)				
Tenancy Condominium Villa Other	991 (22.2) 991 (22.2) 2345 (52.6) 131 (2.9)	583 (22.8) 566 (22.2) 1328 (52.0) 77 (3.0)	408 (21.4) 425 (22.3) 1017 (53.4) 54 (2.8)	0.344
Children				
Living at home *n* (%)	1602 (35.9)	942 (36.9)	660 (34.7)	0.199

**Table 2 healthcare-09-01498-t002:** Means and standard deviations of all variables in the two surveys and *t*-test values.

Dependent Variables				
Independent Variables	Life Satisfaction *t* (4392) = 1.903 *p >* 0.05	Worries *t* (4392) = 4.375 ***	Psychological Distress *t* (4392) = 0.920, *p* > 0.05	View on Public Communication and Activities *t* (4286) = 17.418 ***
	*M (SD)*	*M (SD)*	*M (SD)*	*M (SD)*
**Survey 1**	36.253 (8.373)	10.298 (2.245)	7.151 (2.924)	21.785 (4.577)
**Survey 2**	35.758 (8.674)	10.603 (2.281)	7.236 (3.137)	19.266 (4.806)

*** *p* < 0.001.

**Table 3 healthcare-09-01498-t003:** Beta coefficients of the interaction effects of sex and age on mental health variables.

Dependent Variables				
**Independent Variables**	Life Satisfaction F (5; 4388) = 45.103 ***	Worries F (5; 4326) = 46.286 ***	Psychological Distress F (5; 4388) = 22.567 ***	View on Public Communication and Activities F (5; 4282) = 159.671 ***
	*β*	*β*	*β*	*β*
**Survey**	−0.008	0.145 **	−0.010	−0.363 ***
**Sex**	0.049	0.265 ***	0.056	0.163 **
**Age**	0.182 ***	0.118 ***	0.060	0.099 *
**Survey*sex**	−0.091	−0.100	0.070	0.142 *
**Survey*age**	0.041	−0.093 *	−0.180 ***	0.058

* *p* < 0.05, ** *p* < 0.01, *** *p* < 0.001. *β*, standardized coefficient.

## Data Availability

The study did not report any data.

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
