# Peer review of "Changes in Mental Health and Views on Communication and Activities of Public Institutions among Swedes during the COVID-19 Pandemic—A Cross-Sectional Repeated Measures Design"

_healthcare, 2021, doi:10.3390/healthcare9111498_

Round 1

Reviewer 1 Report

Manuscript: "Changes in Mental Health and Views on Communication and Activities of Public Institutions among Swedes during the COVID-19 Pandemic".

The topic of the manuscript "Changes in Mental Health and Views on Communication and Activities of Public Institutions among Swedes during the COVID-19 Pandemic" provides interesting and current evidence to Healthcare readers.  Some comments are provided to help improve the paper:

Abstract: Should be improved as the manuscript improves with the recommendations following.

Title: According to STROBE guidelines, the study design should be included in the title.

Introduction:

  1. I suggest that authors begin their introduction with a brief contextualization paragraph explaining what COVID-19 is and how the global pandemic it has produced broke out (including time indicators).
  2. It would be interesting if the authors could specify the type of measures (at least name some of them) referred to in line 37.
  3. What is the rate of digital illiteracy in the Swedish population studied? It would be interesting to include this data with the information on line 39, if possible.
  4. The authors can review this website which details publications of a large-scale Spanish study: https://portalcne.isciii.es/cosmo-spain/ I think it may be of interest to better contextualize the problem in this part of the manuscript.
  5. It is suggested that the authors try to improve the thread of their paragraphs, as they change from one idea to another abruptly and this makes it difficult to understand the topic.
  6. The authors should reformulate their objectives so that they are clear and concise. It is suggested that they be expressed according to what they want to explore: association between different variables. Reserve the last paragraph for this purpose, eliminating content that is not a clear objective.

Materials and Methods:

Participants and setting:

  1. The authors should revise their response rate calculation as the figure is not understood being the eligible participants 45000. Again, the survey response rate is not understood 2.
  2. Information was collected from the same participants in survey 1 and survey 2. This is important and it is not clear if they were different participants in survey 2.
  3. The authors should include information on whether informed consent was collected online and whether participants received any compensation for their participation.

Measures:

  1. The authors should provide key information about the online survey, such as total number of items.
  2. What socio-demographic information is collected and how?
  3. What questions are asked in each item? This research needs to be transparent and reproducible and since the survey is ad hoc, it is complex if not all the questions/items asked are detailed.

It is suggested to consult the STROBE guide (https://strobe-statement.org/index.php?id=strobe-home) for observational studies when verifying whether each of the sections of your study includes the minimum necessary information.

Statistical Analysis: This section is quite incomplete and the information on the sample characterization results should be removed from here and shown in the first paragraph of the results.

  1. On the basis of what criteria are regression models fitted for confounding variables?
  2. Is multiple linear regression used? Specify the type of regression.
  3. With which software and model were the analyses performed?

The authors should specify some questions such as, for example, at what value was significance established? by means of what test was the normality of the quantitative variables checked? how were the qualitative variables described in the results? and the quantitative variables?

Results:

1.It is not necessary to show in the text the data of all the categories, for example, of sex. If the one with the highest proportion is shown, the other one is understood or can be consulted in the table. So much information can confuse the reader.

I suggest that the authors carry out a descriptive bivariate analysis and check, if the respondents to survey 1 and 2 are not the same (this is not very clear), whether the respondents to survey 1 and 2 are comparable, i.e. homogeneous in terms of socio-demographic characteristics.

  1. Table 2 is confusing. Please show the variables upside down (dependent variables in the rows and independent variables in the columns), in order to maintain the structure of the table above. Indicate whether and how normality has been checked to show data by the measures used (mean and standard deviation) and show whether or not there are differences by showing the p-values for comparison. Do not use n.s; show all your work.
  2. Table 3 is also confusing. It shows the beta coefficient without its corresponding confidence interval and is titled as "interaction". This is not correct; regression methods show association between variables. The models shown are not understood, if this is what the authors intend to show. It is confusing and they should revise it in detail.
  3. It is not clear which results respond to which study objectives. This needs to be well defined.

Discussion:

The authors should discuss the generalizability of their results based on external validity, as well as clarify their implications for population health. In addition, they should recognize all the limitations of their work, and there are some that have not been recognized, such as the reliability of the measures or those inherent to the cross-sectional study design.

Conclusions:

The authors should synthesize the conclusion, responding only with it, to the study hypotheses.

Reviewer 2 Report

The manuscript has a clear and precise structure. The content is consistent with the title and objective of the study.
It is recommended to describe the applicability of the study and to justify in more detail the incorporation of the variables related to mental health: satisfaction with life, worries, psychological discomfort and views on communication and the activities of public institutions.
Why do these variables determine the mental health of patients?

I agreed to describe how an age-stratified sample was obtained in order to be able to compare the results of the  survey I and the  survey II.

Author Response

Se fil

Reviewer 3 Report

Your work is really interesting and inspiring. Below, some suggestions. 

In Section 2.2 (or 3.1) I suggest that you include some considerations about the sample. In particular, it seems to me that women are overrepresented relative to the general population in Sweden. Could this have affected some interpretations (e.g., lines: 189-190)?

Lines: 184-187: I would suggest using the same scale for Worries (0 - 2) and Psychological Distress (-0.5 - 2.5). Using different scales may cause inappropriate interpretations.

Does the greater life satisfaction of the elderly population (see section 3.2.1) seem to indicate that a greater number of restrictions (lines 37-39) is correlated with greater life satisfaction? If we then look at the Discussion (lines 207-208) the feeling increases that, at least, there is no correlation between psychological distress and pandemic.

And that I think is the point. What is described a little later (235-237) seems to indicate that there is no so-called Swedish model. Alternatively, perhaps that restrictions decided by governments are not the critical factor in explaining increased worry or mental distress in general. If so, I think it is important to make this more explicit.

Author Response

se fil
